



# A climatology of tropical wind shear produced by clustering wind profiles from a climate model

Mark R. Muetzelfeldt[1,2], Robert S. Plant[1], Peter A. Clark[1], Alison J. Stirling[3], and Steven J. Woolnough[1,2]

[1]Department of Meteorology, University of Reading, Reading, United Kingdom
[2]National Centre for Atmospheric Science, University of Reading, Reading, United Kingdom
[3]Met Office, Exeter, United Kingdom

**Correspondence:** Mark R. Muetzelfeldt (mark.muetzelfeldt@reading.ac.uk)

**Abstract.**

A procedure for producing a climatology of tropical wind shear from climate-model output is presented. The procedure is designed to find grid columns in the model where the organization of convection may be present. The climate-model output consists of east–west and north–south wind profiles at 20 equally spaced pressure levels from 1000 hPa to 50 hPa, and the

Convective Available Potential Energy (CAPE) as diagnosed by the model's Convection Parametrization Scheme (CPS). The procedure begins by filtering the wind profiles based on their maximum shear, and on a CAPE threshold of 100 J kg$^{-1}$. The filtered profiles are normalized using the maximum wind speed at each pressure level, and rotated to align the wind at 850 hPa.

From each of the filtered profiles, a sample has been produced with 40 dimensions (20 for each wind direction). The number of dimensions is reduced by using Principal Component Analysis (PCA), where the requirement is that 90 % of the variance

must be explained by the principal components. This requires keeping the first seven leading principal components. The samples, as represented by their principal components, can then be clustered using the K-Means Clustering Algorithm (KMCA). 10 clusters are chosen to represent the samples, and the median of each cluster defines a Representative Wind Profile (RWP) – a profile that represents the shear conditions of the wind profiles produced by the climate model.

The RWPs are analysed, first in terms of their vertical structure, and then in terms of their geographical and temporal

distributions. We find that the RWPs have some features often associated with the organization of convection, such as low-level and mid-level shear. Some of the RWPs can be matched with wind profiles taken from case studies of organization of convection, such as squall lines seen in Tropical Ocean Global Atmosphere, Coupled Atmosphere Ocean Research Experiment (TOGA–COARE). The RWPs' geographical distributions show that each RWP occurs preferentially in certain regions. Six of the RWPs occur preferentially over land, while three occur preferentially over oceans. The temporal distribution of RWPs

shows that they occur preferentially at certain times of the year, with the distributions having mainly one or two modes. Their geographical and temporal distributions are compared with those seen in previous studies of organized convection, and some broad and specific similarities are noted.

By performing the analysis on climate-model output, we lay the foundations for the development of the representation of shear-induced organization in a CPS. This would use the same methodology to diagnose where the organization of convection

occurs, and modify the CPS in an appropriate manner to represent it.





## 1  Introduction

Vertical wind shear is an important factor in the organization of convection in the tropics. From theory, wind shear has been shown to provide the conditions under which squall lines can form (Thorpe et al., 1982; Rotunno et al., 1988), through the interaction of convectively generated cold pools and the environmental shear. Many case studies have highlighted the presence

of wind shear when the convective cloud field has been organized (Barnes and Sieckman, 1984; Cohen et al., 1995; LeMone et al., 1998). Review studies into the organization of convection into squall lines, Mesoscale Convective Systems (MCSs) and Mesoscale Convective Complexes (MCCs) discuss the role of wind shear in the formation of these types of organization (Fritsch and Forbes, 2001; Houze Jr., 2004). Developing a climatology of shear in a climate model can therefore serve as a basis for working out when and where conditions favourable for shear-induced organization of convection will occur.

A climatology is necessarily a simplification that relies on representing the statistical nature of some variable over many years. One question this study sets out to answer is: "how can a climatology of a variable with a large parameter space be created?". In this case, the variable is wind, although the method should be applicable to other such variables. We have designed a method that relies on simplifying the representation of this variable as much as possible, while still retaining the essential features that make the climatology interesting and useful. We make assumptions about which similarities and differences

between the various wind profiles are important, as set out and justified in Sect. 2.2 and following. Through application of our clustering procedure, we reduce the space of all wind profiles down to 10 Representative Wind Profiles (RWPs) that effectively span the space. The 10 RWPs can then be analysed in turn, for example allowing us to see where a specific RWP occurs in space and time.

By producing a climatology of shear in a climate model, we hope to achieve two things. First, we identify regions where

shear-induced organization of convection could be active in the climate model. However, the climate model's convection parametrization scheme does not currently take into account shear, and therefore will not react to take into account the organization of convection that would occur with a given sheared wind profile. Second, we can produce wind profiles and spatio-temporal distributions of shear that can be compared with observations. These comparisons will allow us to build confidence that the climate model is producing realistic wind profiles in sensible places. Taken together, these will point to areas where

the lack of representation of the organization of convection could be having an effect on the climate model's behaviour, as well as providing evidence of the areas where the climate model should be modified to represent organization of convection.

Knowing where the conditions for organization occur in a climate model, it is then possible to compare the climatology of shear in the model to observed distributions of organized convection, such as those in Mohr and Zipser (1996). This helps to





build confidence that the hypothesized link between shear in the model and organized convection holds, and that the model

is producing shear where it should. Further, comparing the wind profiles generated by the model with observed wind profiles from case studies of convective organization also provides a check that the model is producing realistic wind profiles. These comparisons are done in this study, and form the basis of Sect. 4.

Wind shear is detrimental to the formation of tropical cyclones, and so climatologies of wind shear can be linked to the numbers of tropical cyclones in a given year. In Aiyyer and Thorncroft (2006) for example, they look into the climatology

of vertical wind shear over the tropical Atlantic. They define wind shear as a difference in wind speed between 200 hPa and 850 hPa, and develop a climatology over 46 years. Their approach to dealing with the large parameter space is to simplify it dramatically, treating shear as the difference between wind speeds at two levels. However, their motivation is primarily to look for conditions under which tropical cyclones could form, and whether the number of tropical cyclones in a given year could be linked to either El Niño Southern Oscillation (ENSO) or precipitation in the Sahel region, and so is quite different from the

focus of this study.

Houchi et al. (2010) developed a global climatology of mean wind and wind shear profiles, from the surface up to 30 km. They do this over four zonal bands representative of the tropics, the Northern Hemisphere (NH) subtropics, the NH mid-latitudes and the NH polar region. They compare radiosonde data with co-located ECMWF operational forecasts, finding that the model produces realistic wind profiles, but underestimates the shear due to it not reproducing the fine structure of the wind

profiles. The motivation for their work was to choose optimal vertical bins for the Atmospheric Dynamics Mission Aeolus satellite, and so opportunities for comparison against the work presented here are limited as they are focused on answering different questions which means their analysis cannot easily be compared with ours.

Chen et al. (2017) investigated the link between large-scale predictor variables and large (rain area $> 10000$ km$^2$) precipitation features. They used data from the Tropical Rainfall Measuring Mission (TRMM) satellite to provide information about

the distribution of MCSs, and ERA-Interim to obtain information about the large-scale environment. They investigated the occurrence of MCSs, conditional on several predictor variables. They found the size of precipitating systems is best predicted by total precipitable water vapour, relative humidity and shallow vertical wind shear. Deep vertical wind shear is shown to be a weak predictor at best.

Although relatively few climatologies of wind shear are available, several climatologies have been produced for the or-

ganization of convection (Mohr and Zipser, 1996; Laing and Fritsch, 1997; Tan et al., 2015; Huang et al., 2018). Some of these provide geographical distributions of where various types of organized convective systems are likely to occur (Mohr and Zipser, 1996; Huang et al., 2018). These studies tend to be based on satellite observations though, and so do not relate the organization to the wind shear, which in many cases may be responsible for creating the organization. Thus, comparison of the shear climatology produced here with those existing climatologies should increase confidence both that the climate model is

producing the conditions for organization in the right locations, and that the organization is being influenced by the sheared wind profile.

Many case studies have looked into specific events of organized convection (e.g., Houze Jr., 1977; Jorgensen et al., 1997), or regions where there is a typical mode of organization (e.g., Barnes and Sieckman, 1984; Cohen et al., 1995). Some of





these studies provide hodographs or wind profiles; these can be compared to the RWPs produced here to look for similarities

between them from certain regions. Performing this comparison also provides a check that the model is producing realistic wind profiles in the correct regions.

Producing a climatology of wind profiles can give insights into where and when the organization of convection is likely to occur in a climate model. The majority of Convection Parametrization Schemes (CPSs) do not currently take into account the organization of convection. We envisage that the diagnosis of wind profiles which are associated with the organization

of convection developed here, in combination with a characterization of the convective response to them performed in future work, could lead to the development of a "shear-aware" CPS which can represent the shear-induced organization of convection. The RWPs produced here could form the basis of such a study, which would analyse the degree of organization expected from each RWP. This is explored further in Sect. 4.5.

The rest of this study is structured as follows. In Sect. 2, we provide information about the climate model used to generate a

suitable dataset of wind profiles, as well as providing an outline and then detailing the clustering procedure used to turn these profiles into a set of RWPs. In Sect. 3, we analyse the results of the individual RWPs, and analyse the spatial and temporal distribution of the RWPs. In Sect. 4, we discuss our results in relation to previous studies of organization, and outline some ideas for future work. In Sect. 5, a brief summary of the major results is given.

## 2 Methods

### 2.1 Climate model

The climate model that is used is the United Kingdom Met Office's Unified Model (UM), version 10.9. It is run using the standard Global Atmosphere 7.0 (GA7.0) settings, as described in depth in Walters et al. (2019). The interested reader is directed to that paper for the full details. It is an atmosphere-only model, using prescribed sea surface temperatures. It uses a version of the Gregory–Rowntree convection scheme (Gregory and Rowntree, 1990), which is a mass-flux scheme that, in

its current implementation in the UM, uses a Convective Available Potential Energy (CAPE) closure. Here, it is run with an N96 resolution, which corresponds to a zonal grid spacing of 209 km at the equator. It is run for five years, from September 1988 to August 1993 using a 360 day calendar. Running for five years allows us to sample inter-annual variation. The period covers both a positive and negative phase of ENSO, which means that it should be representative of the typical conditions a climate model can produce. East–west ($u$) and north–south ($v$) winds are output on 20 pressure levels from 1000 hPa to 50

hPa with a resolution of 50 hPa, and are output every 6 hours. CAPE is also output every 6 hours; it is calculated by the model's convection scheme using an undilute parcel ascent and output as a diagnostic field. Profiles are only considered in the tropics, defined as being between 23.75° N and 23.75° S.





## 2.2 Overview of clustering procedure used to generate the representative wind profiles

The climate model used here has 70 vertical levels, with an east–west and north–south component of the wind at each level.
This leads to a large parameter space for wind profiles. The problem of producing a climatology of these profiles then becomes one of choosing how to reduce the complexity of the parameter space, while still maintaining the essential features that link a given group of profiles together. To do this, we have made some simplifying assumptions about what aspects of the profiles will provide useful information about shear-induced organization.

First, we only consider wind values over the depth of the troposphere, from 1000 hPa to 50 hPa in steps of 50 hPa. Each
wind profile, or sample, then has 40 dimensions. We are interested in grid columns in which shear-induced organization of convection is active, hence we filter the profiles based on their grid column's CAPE and shear. We also recognize that the low-level and mid-level tropospheric shears are more important for the organization of convection by weighting the contribution to the analysis from the lower troposphere (up to 500 hPa) more highly. This is necessary to stop the higher-level jets dominating the analysis and clustering procedure. Although some studies have shown that shear at higher levels can be important for
organization (e.g., Chen et al., 2015), focusing on the lower troposphere can be justified from the results of theoretical studies such as those of Rotunno et al. (1988); Thorpe et al. (1982), and observational studies such as LeMone et al. (1998). Likewise, Chen et al. (2017) showed that deep shear was a poor predictor for MCS activity, whereas low-level shear was a better predictor. Second, we do not expect the relative rotation of the wind profiles to have a large effect on the organization that they induce. We therefore choose to neglect the relative rotation of the wind profiles.

The dimensionality of the problem can be further reduced by using Principal Component Analysis (PCA) to extract principal components that capture most of the variance of the samples with fewer dimensions. Then the samples, as represented by their principal components, can be grouped together using a clustering algorithm. Clustering is a form of unsupervised machine learning. It groups similar samples in a dataset together, based on how close they are to each other. In this study we use the K-Means Clustering Algorithm (KMCA), as it provides a simple and efficient way of clustering like samples together. Once the
samples have been clustered, the median of each cluster is referred to as an RWP. The clustering is done entirely on the values from one grid column, so this technique is a data-driven way of grouping together like grid columns. It could be performed on any set of values from a grid column, and a similar method is used by Hoffman et al. (2005) to group together land grid cells in a climate model based on which plant types were present.

For both the PCA and KMCA algorithms, we use the implementations as provided in the scikit-learn Python package
(Pedregosa et al., 2011).

The clustering procedure is shown schematically in Fig. 1, and the details for each of the steps follow.





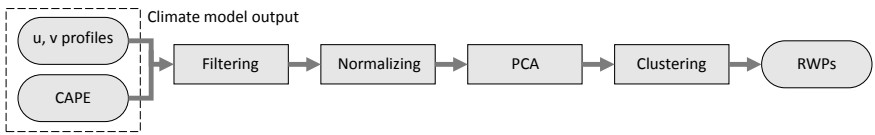

**Figure 1.** Schematic of clustering procedure. The four processing steps are shown as rectangles, and the input and output as rounded rectangles.




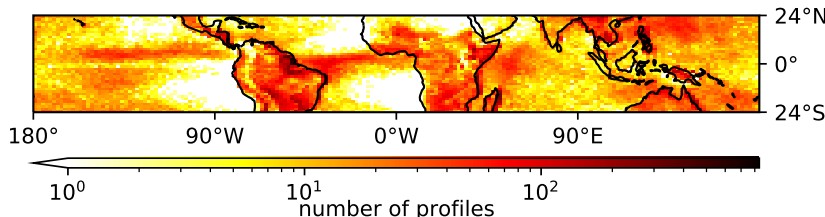

**Figure 2.** Heatmap of grid columns that have been filtered based on CAPE and maximum shear. The heatmap shows the number of profiles in each grid column that have been kept after applying both filters to the output of the five years of the simulation.

## 2.3 Filtering

Profiles are filtered on two criteria. The filters reduce the number of profiles, excluding ones that either are not likely to produce convective activity of sufficient intensity to produce organization or are not profiles with large amounts of shear. The filters are applied independently, so their order of application makes no difference. The filters in use are as follows.

1. Exclude grid points where CAPE $< 100 \, \mathrm{J \, kg^{-1}}$. This is done to restrict the profiles to ones where convection is likely to be active. Using values above $100 \, \mathrm{J \, kg^{-1}}$ ensures that the convection is more likely to be vigorous, such as when MCSs have been observed (see e.g. Betts et al. (1976), Table 2, where the minimum value of environmental CAPE associated with squall lines is $123 \, \mathrm{J \, kg^{-1}}$);

2. Exclude grid points where the maximum shear in the profile is less than the 75th percentile of profiles from across the tropics. The shear is calculated at the midpoint between each pressure level, and only the shear values up to the level of $500 \, \mathrm{hPa}$ are taken into account to focus on the lower troposphere.

Filtering on CAPE keeps 5.1 % of the profiles, while the maximum shear filter keeps 25 % of profiles by construction. Both filters combined, the intersection of the two sets of filtered profiles, keep 0.30 % of profiles. Note, as each filter is independent, the combined percentage of profiles need not be the same as the product of the percentage each filter keeps. It is worth noting that fewer of the profiles are kept than would be seen by the product of the percentages; this implies both that in general profiles with strong shear have less CAPE than average, and that profiles with high CAPE have weaker shear than average.

Fig. 2 shows the spatial distribution of filtered profiles across the tropical belt. Some features that stand out are the reduced number of profiles over the South Atlantic Gyre and South Pacific Gyre, as well as enhanced bands across the equator in the Atlantic, and just north of the equator in the Pacific. These are similar to features seen in global precipitation climatologies, see e.g. Fig. 4 in Adler et al. (2003), which describes the Global Precipitation Climatology Project (GPCP). However, there are some differences with global precipitation climatologies, such as over the western Sahara and over the western Indian Ocean. There is also a high density of profiles in the north-west Pacific, extending further north than is seen in GPCP. This may be due to the fact we are using a threshold based on CAPE, and not based on precipitation, to filter the profiles.




## 2.4 Normalization

It is necessary to pre-process the data before performing the principal component analysis and the clustering, normalizing the magnitude and the rotation so that differences that are most important between the profiles for this analysis are brought to the fore.

### 2.4.1 Normalize magnitude

Normalization of the magnitude of the samples involves normalizing each profile by the maximum magnitude of the wind at each pressure level (i.e. $\sqrt{u^2 + v^2}$), restricting the normalized magnitude at each pressure level to between 0 and 1. This is done to ensure that differences in the profiles at each pressure level each contribute the same amount to the distance measure used by PCA and the KMCA.

To favour the lower troposphere, an extra factor is applied to the normalization above 500 hPa: the normalized values above this height are reduced by a factor of four. This choice of parameter has the effect of reducing the contribution of differences in the upper troposphere when applying the clustering. However, information about what happens in the upper troposphere is retained, making it possible to determine the shape of the profiles up to a maximum height of 50 hPa. We discuss the sensitivity to this choice of the parameter in Sect. 4.3, finding that the results are not overly sensitive to it.

### 2.4.2 Normalize rotation

Normalization of the rotation is applied to treat profiles that share rotational symmetry in the same way. It is done by using the wind vector at 850 hPa to define a rotation angle. All profiles are then rotated so that this angle is zero, i.e. all the profiles are aligned in the same direction at 850 hPa. A small number of profiles (2.9 %) have a wind speed less than $1 \, \mathrm{m\,s^{-1}}$ at 850 hPa. These profiles are included but it should be noted that they may influence results. After applying this normalization, $u'$ refers to the direction aligned with the wind vector at 850 hPa, and $v'$ is the orthogonal direction to this. The reason for applying this normalization is that in the tropics, it makes little difference whether a profile has, for example, unidirectional shear in the zonal or meridional direction. A similar point stands for all profiles, such as profiles that veer or back with height.

## 2.5 Principal component analysis

PCA is used to reduce the number of dimensions of each sample, by projecting each sample onto a truncated set of principal components. PCA is a process that finds orthogonal, unit length principal components of a dataset that are linear combinations of the original axes. It does this in such a way that the first principal component accounts for the largest possible amount of variance in the underlying dataset. The second principal component is orthogonal to the first, and accounts for as much of the remaining variance as possible in the dataset, and so on. PCA can be used to reduce the number of dimensions of a dataset in a way the keeps the maximum possible variance for a given number of dimensions, by truncating the number of principal components used.



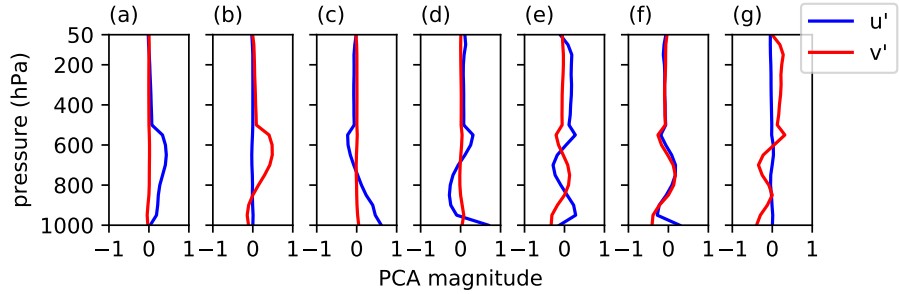

**Figure 3.** The first to the seventh principal component, shown in (a) to (g) respectively, as profiles of magnitudes between -1 and 1, for both $u'$ and $v'$.

The algorithm for PCA works by centring the dataset on the mean for each of its dimensions, and then calculating the covariance matrix for this dataset. The principal components can then be taken by finding the eigenvectors of the covariance matrix, with the first component corresponding to the eigenvector with the highest eigenvalue, and so on.

The number of dimensions of each sample in the original dataset is 40 (20 pressure levels for $u$ and $v$). We have chosen to keep as many principal components as are required to explain 90 % of the variance, which for this dataset is seven. The seven principal components are shown in Fig. 3. All principal components show lower magnitudes above 500 hPa. This is due to the higher weighting of the lower troposphere, as described in Sect. 2.4.1. The first four principal components individually show little turning with height, though a linear combination of them could be used to represent a profile with some turning. The first principal component describes shear parallel to the wind at 850 hPa, and the second describes shear perpendicular to the wind at 850 hPa. The first principal component describes wind increasing with height and parallel to the wind at 850 hPa, whereas the third describes wind decreasing with height in this direction. Beyond the third principal component, the structure of the remaining ones becomes increasingly complex.

The leading principal components bear some resemblance to sine and cosine functions, suggesting that using Fourier modes could be used as an alternative way of representing the profiles with a reduced number of dimensions. This would not work so well with the favouring of the lower troposphere though, and may also require more dimensions to represent an equivalent amount of the variance as well.





## 2.6 K-means clustering

The KMCA splits a number of samples into clusters based on how similar the samples are to other samples. It does this in a way which minimizes the within-cluster variance. The algorithm used here is Lloyd's algorithm (Lloyd, 1982), which is computationally efficient but not guaranteed to find a global minimum for the within-cluster variance. It starts by randomly assigning samples to clusters. Then it calculates the mean of each cluster, and re-assigns samples to new clusters based on which of the cluster means each sample is nearest to under a Euclidean distance metric. This last step is repeated until there is very little movement (less than some predefined threshold) in the cluster means, when the algorithm terminates.

The number of clusters to use is not a priori obvious. We have the competing requirements that we want few enough RWPs that we can analyse where each one comes from without being overwhelmed with data, and we want each RWP to be as representative as possible of its cluster of profiles. We pick 10 clusters as a pragmatic number of clusters to use, being large enough to span the wind profile space, and small enough that sensible analysis of each cluster is possible. As discussed in Sect. 4.4, we investigate using different numbers of clusters, finding that no value for the number of clusters produces an optimal outcome in terms of identifying inherent clusters in the data.

We ran the algorithm five times with different random seeds. The results each time were nearly indistinguishable in terms of the RWPs that were produced, although the particular cluster label would be different for each of the different initial seeds. This indicates that the algorithm is unlikely to be getting stuck in a local minimum.

## 3 Results

### 3.1 Representative wind profiles

The 10 RWPs are shown in Fig. 4, along with the 10th and 90th percentiles at each pressure level. They have been denormalized with respect to magnitude (but not to rotation), hence their wind speed is shown in $\mathrm{m\,s^{-1}}$ on the abscissa. It should be noted that there is no spread in the $v'$ profile at $850\,\mathrm{hPa}$; this is due to the normalization of the rotation to align all profiles in the $u'$ direction. The spread of the percentiles increases going higher into the atmosphere. This is caused by the contribution of the lower troposphere being favoured, as described in Sect. 2.4.1.

Five profiles show low-level shear: C2, C5, C6, C7 and C10 have maximum differences in wind speed of greater than 8 $\mathrm{m\,s^{-1}}$ between $1000\,\mathrm{hPa}$ and $800\,\mathrm{hPa}$. C6 shows particularly strong difference in wind speed of $24\,\mathrm{m\,s^{-1}}$ between $1000\,\mathrm{hPa}$ and $800\,\mathrm{hPa}$. Given that low-level shear is so important for the organization of convection, this implies that these RWPs might indicate where organized systems are likely to occur. Profiles C2, C10 and C4 respectively look like scaled versions of a similar profile, with C2 having the strongest winds and C4 the weakest.

Profiles C1, C3, C8 and C9 contain mid-level shear, with a maximum difference in wind speeds of greater than 8 $\mathrm{m\,s^{-1}}$ between $800\,\mathrm{hPa}$ to $500\,\mathrm{hPa}$. From the evidence in observational studies (LeMone et al., 1998), and modelling studies (Robe and Emanuel, 2001), this could be important in determining the organization, particularly if there is weak low-level shear, as is the case for RWPs C1, C3 and C8.



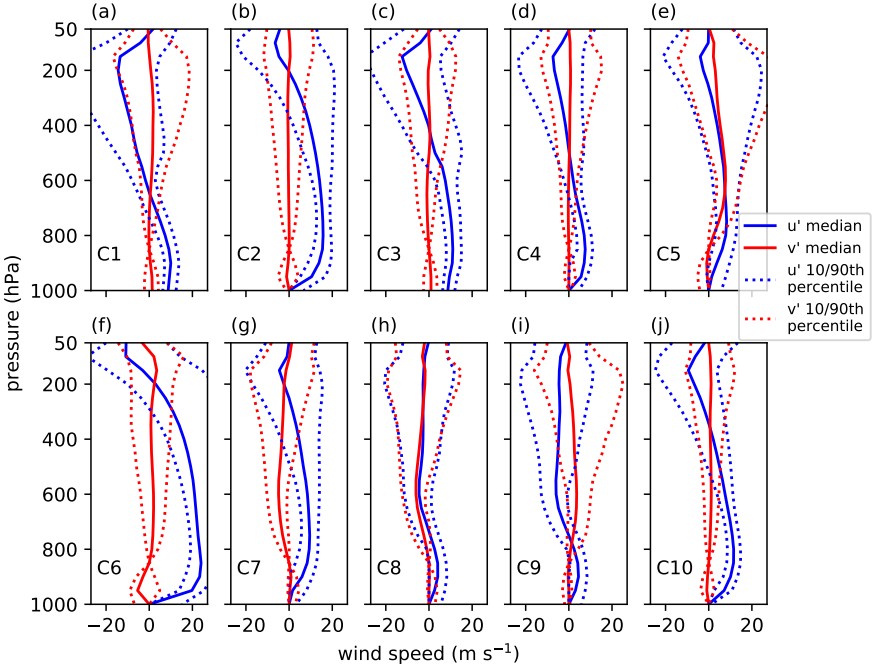

**Figure 4.** $u'$ and $v'$ wind profiles of the median of the 10 RWP clusters, with (a) showing cluster 1 (C1), (b) showing cluster 2 (C2) and so on. They have been denormalized with respect to magnitude. The $10^{th}$ and $90^{th}$ percentiles are also shown.

As is more clearly visible in the hodographs in Fig. 5, profiles C5, C6 and C7 all show some turning with height. C7 is seen to be backing with height, while C5 and C6 are seen to be veering.





## 3.2 Geographical distribution of representative wind profiles

Fig. 5 shows hodographs of the 10 RWPs, wind roses of the distribution of the rotation of the profiles, and heatmaps of the geographical distribution of each RWP. From the geographical distributions, it is evident that the various RWPs show clear spatial patterns, with some RWPs showing more activity over specific regions of the globe, for example C6 occurs predominantly over the north-west Pacific. As can be seen in Table 1, some RWPs occur almost exclusively over the oceans, the C1, C3 and C6 occurrences are 90.8 %, 93.6 % and 85.8 % over the oceans respectively. This in contrast to RWPs C4, C5, C7, C8, C9 and C10, that occur between 70 % to 82 % over land. RWP C2 is the only RWP that is roughly split between ocean and land.



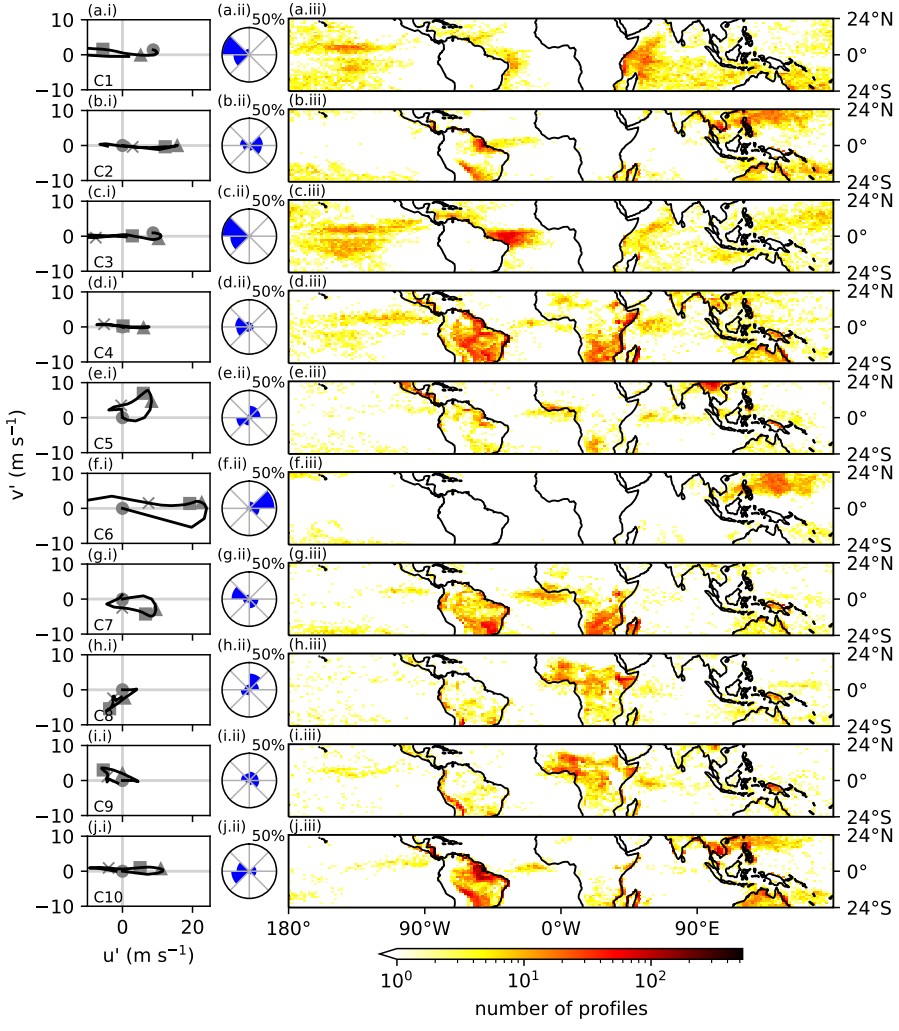

**Figure 5.** Left column: hodographs of the 10 RWP clusters, showing the $u'$ and $v'$ winds as in Fig. 4. The circle, square, triangle and cross show levels 1000 hPa, 750 hPa, 500 hPa, 250 hPa respectively. Centre column: Wind rose of distributions of the wind direction at 850 hPa. (Note that these show the direction in which the wind at 850 hPa is blowing, i.e. for C3 the 850 hPa wind is to the west, or easterly, as can be seen from (c.ii).) Right column: heatmaps of the distributions of each RWP across the tropics, showing the total number of profiles for each RWP at each grid column.





**Table 1.** RWPs, showing the percentage that occur over land and ocean, and the total number of profiles in each RWP.

| RWP | Land % | Ocean % | Number of profiles |
|---|---|---|---|
| C1 | 9.23 | 90.77 | 14356 |
| C2 | 53.70 | 46.30 | 15111 |
| C3 | 6.42 | 93.58 | 15844 |
| C4 | 75.31 | 24.69 | 29633 |
| C5 | 70.44 | 29.56 | 11023 |
| C6 | 14.18 | 85.82 | 5579 |
| C7 | 74.46 | 25.54 | 16908 |
| C8 | 76.17 | 23.83 | 11437 |
| C9 | 75.57 | 24.43 | 11982 |
| C10 | 81.26 | 18.74 | 27249 |

Table 1 also shows that there are different numbers of profiles in each RWP. C6 has the fewest with 5579, and C4 has the most with 29633, approximately five times as many. The mean number of profiles in an RWP is 15912.

RWP C6 is the most localized of all the RWPs; as noted above, almost all occurrences are over the north-west Pacific. Figure 6 shows the annual cycle of RWP occurrences. For C6, most of the activity happens in July and August, meaning that it is localized in time as well. It is an interesting profile in terms of its vertical structure, having strong low-level shear (Fig. 4). The fact that it is spatially and temporally localized fits with it having the fewest number of profiles.

From Fig. 4, RWPs C2, C10 and C4 were identified as having similar vertical structures. C2 and C10 show broadly similar
geographical distributions: they both occur across South America and the north-west and south-west Pacific (Fig. 5). However, C2 is more active in the north-west Pacific, and shows activity in the south-east Pacific whereas C10 shows very little. Looking at the wind roses for C2 and C10 shows that the winds at 850 hPa are orientated in different directions, being predominantly eastwards for C2 and westwards for C10. C4 shows a much greater difference in spatial distribution from C2 and C10; there is considerable activity over southern Africa and over the equatorial Pacific. Like C10, the 850 hPa winds are orientated
mainly westwards. Therefore, despite their broad similarity in terms of their vertical structure, these RWPs are seen to have distinguishing features in terms of how they are distributed across the tropics and in their orientation.

RWP C1 occurs most often over the west Indian Ocean (Fig. 5), and also has high levels of activity over the west Atlantic, and over the Pacific. The activity over the Indian Ocean comes mainly from the months of JJA (see Fig. S4 in supplementary material), which corresponds to the peak seen in its temporal distribution in Fig. 6. The activity in the Pacific shows seasonal
dependence, with activity being highest in the winter hemisphere (Fig. 6). RWP C3 is prominent over the equatorial Atlantic, as well as showing some activity over the Pacific and Indian Oceans. C5 shows little spatial localization. C7 occurs mainly over land: especially over South America and southern Africa. There is also some signal over the east equatorial Atlantic. C8 is prominent over Africa, and C9 shows signals over Africa and South America, particularly along the west coast which is perhaps related to the position of the Andes.





From the wind roses of orientation (Fig. 5), four RWPs are seen to be almost exclusively orientated in a particular direction: C1, C3, C4 and C6. Others are predominantly in one direction: C2, C7, C8 and C10. C5 appears to be bimodal, either directed north-east or west-south-west. C9 has a broad spread of orientations in its distribution. However, it has weak winds at 850 $\mathrm{hPa}$ (see Fig. 4), and this may lead to there being more chance of it having different orientations. From the hodographs (Fig. 5), half of the RWPs do not show much turning with height, for example C1, C2, C3, C4 and C10. C6 shows significant turning

from near the surface up to around 500 $\mathrm{hPa}$, and C5 and C7 both show similar magnitudes of turning in both the $u'$ and $v'$ directions.



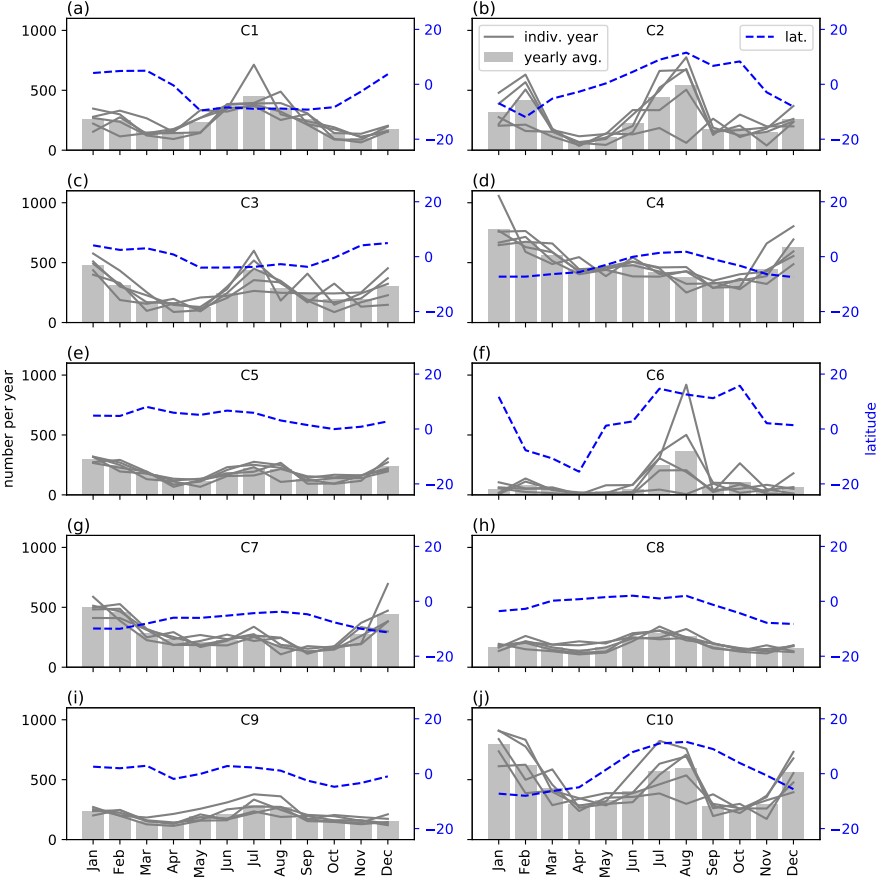

**Figure 6.** Seasonal variation of the 10 RWPs, showing the yearly average, each year individually and the mean latitude of occurrence.

## 3.3 Seasonal variation of RWPs

In Fig. 6, the seasonal variation of each of the RWPs is shown. Additionally, the mean latitude of all the profiles that make up each RWP is shown. In most cases, there is little variation from year to year, and C2, C6 and C10 are the only RWPs that show much interannual variability. C2 shows variability over February, and all three show variability over July and August. In all of these cases, variability occurs when there is a peak in activity.

Several RWPs show a bimodal seasonal distribution – C1, C2, C3, C5, C9 and C10. For C1, C2 and C10, there is a clear latitudinal dependence on the month as well, indicating that the cause for the bimodal seasonal distribution is that the RWP is active in either the summer or the winter hemisphere. For C2 and C10, there is more activity in the summer hemisphere, whereas for C1 the opposite is true. C4 shows strong activity all year round, peaking in January and having a slight increase in activity in the summer hemisphere. C6 shows a very strong increase in July and August, with high variability across years





as well (there is little activity in September 1988–August 1989). The other RWPs, C7 and C8, show weaker seasonal and latitudinal dependence.





## 4 Discussion

The analysis so far can be used to shed light on where and when particular shear conditions are active in a climate model. In this section, we draw out the link between shear and the organization of convection by associating specific features seen in our analysis with similar features seen in studies of the spatial distribution of MCSs. We also examine whether the RWPs we have identified are similar to wind profiles found in case studies of organized convection.

### 4.1 Comparison with studies of the spatial distribution of MCSs

Mohr and Zipser (1996) use satellite observations of the 85 GHz frequency band to detect tropical MCSs over January, April, July and October 1993. The 85 GHz band picks up the ice scattering signature, and so is well suited to detecting the large cirrus shields associated with MCSs. They define an MCS as an area with brightness temperature below 250 K of at least 2000 km$^2$, which contains a minimum brightness temperature of below 225 K. From this, they produce a distribution of MCSs between 35° N and 35° S. They find that there is a difference in distribution between continental and oceanic MCSs, with continental

MCSs being 60 % more frequent at sunset and oceanic MCSs being 35 % more frequent at sunrise. We can compare our distributions of RWPs in Fig. 5 with their distributions of MCSs, to determine whether our climate model is producing sheared environments in the expected regions of the tropics.

Mohr and Zipser (1996) present two distributions for each month: one for the sunrise pass and one for the sunset (their Figs. 4–7). These can be compared with the filtered profiles (Fig. 2) and the geographical locations of the RWPs (Fig. 5). The first

thing to note is that there is a broad agreement between regions where there is MCS activity and where we see activity of the RWPs. High MCS activity is seen over Africa, South America and the tropical oceans. This corresponds to high levels of activity of RWPs over Africa (C4, C7, C8 and C9), South America (C2, C4, C7, C10) and the tropical oceans (C1, C3 and C4). Also, in their study, they see reduced activity over the South Atlantic Gyre and South Pacific Gyre, which is also seen in Fig. 2 and Fig. 5.

A latitudinal variation of activity over Africa and South America is seen in their Figs. 4–7. This matches well with the RWP geographical distributions when each RWP is considered separately for the seasons DJF, MAM, JJA, SON respectively (see supplementary material, Figs. S2–S5). Also, there is a peak in activity over Australia of C2, C4 and C10 over DJF (see supplementary material, Fig. S2), corresponding to enhanced activity in their Fig. 4 (b) in January.

RWP C6 was shown to be well localized in time and space, occurring predominantly over the north-western Pacific in the

months of July and August (Figs. 5 and 6). In Fig. 5 (a) and (b) of Mohr and Zipser (1996), increased activity is seen over a region defined by 25° N to 5° N and 120° E to 150° E. From their other figures, MCS activity in this region is seen to be much reduced in January and April, and slightly reduced in October. Given that C6 has strong low-level shear, it is not unreasonable to expect it to cause organization of convection. We can therefore tentatively link the wind profiles in our climate model with the organization seen in this region and season. We note that other RWPs are active in this region at the same time, namely C2

and C10, and these too could be helping to produce organization of convection in this region, due to their low-level shear.





There are however some regions where RWPs occur, and no corresponding MCSs are observed. RWPs C1, C3, and C4 occur over the western Indian Ocean, whereas little organization of convection is seen in any month over this region. Similarly, RWPs C4 and C8 show activity over the Horn of Africa, and no MCSs are detected there. It has been documented that there is too much precipitation over the western Indian Ocean in the UM in previous studies (e.g., Bush et al., 2015), and this bias could be responsible for the increased RWP activity over the same region. This could explain, or could be caused by, the same bias that causes there to be too much CAPE in this region, which would affect our filtering step.

Comparison with other studies that have produced climatologies of organization also provides some useful information. Huang et al. (2018) track MCSs using an algorithm that combines a Kalman filter with an area-overlapping method, using satellite infrared data. Their MCS definition criteria include a requirement that the area of cloud brightness temperature that is 233 K or under must be over 5000 km$^2$, and so is stricter than that of Mohr and Zipser (1996). They produce a figure that can be compared with our Fig. 2: their Fig. 4. However, they note the similarity between their Fig. 4 (a) and GPCP. We have already discussed the differences between our Fig. 2 and GPCP in Sect. 2.3, and so it is not surprising that very similar remarks apply between our Fig. 2 and their Fig. 4 (a). In Tan et al. (2015), their Extended Data Figure 1 (a) shows a geographical distribution of "Mesoscale organized deep convection", which we will take to be similar to the definition of MCSs. Again, the similarities with GPCP and our own RWP results are apparent.

## 4.2 Comparison with wind profiles from regional case studies

Numerous case studies of the modes of organization of convection present in particular regions of the tropics have been carried out. For example, in Betts et al. (1976), the properties of six squall lines over Venezuela are investigated over the summer of 1972, showing that a theoretical model provides a good match with the observed squall lines. In Cohen et al. (1995), a selection of squall lines over the Amazon during April and May, 1987, is examined. They link the squall line formation to the environmental wind shear, and present hodographs associated with long-lived squall lines. LeMone et al. (1998) look at squall lines over the tropical Pacific as part of the Tropical Ocean Global Atmosphere, Coupled Atmosphere Ocean Research Experiment (TOGA–COARE) over 1992 and 1993. They show many hodographs, which are used to back up their main findings that organization of convection is primarily controlled by low-level and mid-level shear. We find that we can match some of our RWPs with profiles from these studies.

Betts et al. (1976) show a composite wind profile from four squall lines in their Fig. 5. We are interested in the inflow, which we will take as being the environmental conditions. Their profile shares many features with C10. It has near zero absolute wind at the surface, reaches a maximum wind of $10\,\mathrm{m\,s^{-1}}$ at 700 hPa, reducing to zero again by 300 hPa. Compared to C10, this means its maximum wind is of a similar magnitude, although it occurs slightly higher up. The geographical distribution of C10 is consistent with activity over Venezuela; it shows activity over Venezuela and much of South America. Its orientation is consistent with the profile from Betts et al. (1976) as well, with the wind at 850 hPa being oriented primarily to the west.

In Cohen et al. (1995), Fig. 2 (c) shows a hodograph from a case where a squall line was formed. The hodograph matches C10 reasonably well. Their hodograph has more turning in the lower troposphere, but the maximum deviation from a linearly aligned wind profile is $2\,\mathrm{m\,s^{-1}}$. There is more surface wind than C10 as well, as their profile shows approximately $5\,\mathrm{m\,s^{-1}}$





of surface wind whereas C10 has no surface wind. However, it shows a reversal of wind at 800 hPa to 850 hPa, which is
similar to the level of reversal of C10 (from Fig. 4). The magnitude of wind is $14\,\mathrm{m\,s^{-1}}$ at this level, which is higher than the
magnitude in C10 of $10\,\mathrm{m\,s^{-1}}$. The $u$ wind goes to zero by 350 hPa, very close to where the $u$ wind in C10 is zero. From Fig.
5, C10 can be seen to be active over South America, and the study is looking at Amazonian squall lines. From the wind rose
distribution, the 850 hPa wind in C10 can be seen to be easterly, which matches the profile from the paper.

In LeMone et al. (1998), the profile in the hodograph in Fig. 2 (upper) again matches C10 very well. This time, the surface
wind is approximately zero, there is little deviation from a linearly aligned wind profile, the maximum magnitude is around 10
$\mathrm{m\,s^{-1}}$ and the $u$ wind goes to zero at 400 hPa. The profile was taken from 7.7° S, 158.8° E, which places it in the south-west
Pacific. From Fig. 5, C10 can be seen to show some activity here, although in general C10's activity is further south. The
LeMone et al. (1998) profile shows a westerly wind at 800 hPa, and from the wind rose distribution, some of the C10 profiles
show a westerly wind. The profile in the lower panel of Fig. 2 from LeMone et al. (1998) is seen to share some features with
C5, although this time the match up is not as strong. There is a degree of backing with height, which matches C5, although
there are stronger surface winds and winds at 850 hPa in the profile from the paper. The profile is from 10.3° S, 157.9° E, and
C5 is active in this region, and the wind rose distribution indicates that the orientation of C5 is consistent with the orientation
of the profile. Finally, the profile in Fig. 8 (lower) from LeMone et al. (1998) shares a similar form to C3. C3 is active in the
correct regions, but its wind rose distribution (Fig. 5) shows no sign of westerly aligned profiles.

Taken together, these comparisons show that some of the RWPs are consistent with organized convection and occur in similar
regions to those observed in previous studies. As noted earlier, C2, C10 and C4 are quite similar in form. Given that C10 was
matched with three of the profiles from case studies, it is not unreasonable to expect that these profiles will be associated with
organization. That we can see correspondence between the RWPs and profiles in case studies suggests that the climate model is
producing realistic wind profiles in places that are associated with the organization of convection. This lends weight to the idea
that we can use information on the wind profile distributions to find areas where the organization of convection is prevalent.

We note here that there are many studies of organized convection that we have not compared against, and many profiles both
from these studies and studies of individual squall lines for which it may not be possible to find a good match. However, we
wish here to show that we are capturing some of the wind shear structure that is associated with the organization of convection,
without necessarily claiming that we can match every profile.

### 4.3 Sensitivity of results to choices of parameters

There are several parameters that have been chosen as part of the RWP generation procedure. These have been chosen in order
to pick out various features in the underlying data. For example, the CAPE threshold selects profiles where organization of
convection is likely to be active. It is useful to conduct some sensitivity analyses to make sure that the conclusions that would
be drawn using other reasonable parameter choices are broadly the same. Specifically, we varied:

- CAPE threshold: $75\,\mathrm{J\,kg^{-1}}$ and $125\,\mathrm{J\,kg^{-1}}$;

- maximum shear percentile: 65th and 85th; and





  – lower troposphere favour factor: 3 and 5 times.

Running with these different values makes some difference (see supplementary material Figs. S11–S16), but the over-
all conclusions would be the same. The differences are most notable in that more/fewer profiles are in each RWP for de-
creased/increased CAPE threshold and maximum shear percentile. In all cases, there is a one-to-one correspondence between
the RWPs in the control set and the sets produced with modified parameters. Changing the lower troposphere factor has an
effect on the number of principal components needed to represent 90 % of the variance: using values of three and five needs
eight and six principal components respectively. However, it does not substantially affect the RWPs that are generated.

**4.4   Use of clustering**

The use of clustering as the final step in the procedure partitions the space of all wind profiles into $N$ clusters. However, the
question of how to pick $N$ is still open. There are methods for estimating what value to use, such as the elbow method, which
runs the algorithm for different values of $N$, looking for an identifiable kink in the resulting within-cluster variance score.
However, when we ran with 5–20 clusters, no obvious kink could be seen (see Fig. S17 in supplementary material), indicating
that there was no particular number of clusters that produced an optimal outcome. This is perhaps to be expected; we cluster
the data to find coherent groups within it, but the cutoff between one group and another is in a sense arbitrary. This is due
to our underlying dataset not exhibiting strong inherent clustering. Our requirements are rather that we need to have enough
clusters so that each RWP can be said to be representative of all the member profiles, and few enough that we can analyse each
of the clusters individually and compare all the clusters with observations. We therefore take $N = 10$ as a pragmatic choice,
noting that the spread between the 10[th] and 90[th] percentiles in Fig. 4 is not too large and that we can say something physically
meaningful about each of the clusters.

As noted earlier, the fact that we end up with nearly indistinguishable results running with different seeds means that the
algorithm is unlikely to be getting trapped in a local minimum. It also points to the fact that there is some bunching of the
samples together.

**4.5   Extensions**

Some extensions to the work that we have presented here naturally suggest themselves. We have produced a climatology
of shear for a climate model. We do this so that the model's reproduction of the shear environment for convection can be
understood, and partially evaluated against observations. However, using a climate model makes comparison with observations
more difficult, as multiple aspects of the analysis are being tested at once. First, the climate model must be correctly producing
wind profiles that are similar to those seen in observations. Second, the model must produce these in the correct place. Third,
our analysis must correctly identify the profiles as RWPs, with enough discriminatory ability to distinguish between different
RWPs but not so much that we are overwhelmed by the number of RWPs to analyse.

One way of performing a similar analysis in a manner that was more tied to the observed state of the atmosphere would
be to use a reanalysis product, such as ERA-20C (Poli et al., 2016) or 20[th] Century Reanalysis Project (Compo et al., 2011).





Due to these being combinations of a model and assimilated observations, they may well produce wind profiles that are both more realistic and closer in location than those of a climate model. With this analysis, the comparison with observation would provide stronger evidence that the shear climatology was indeed influencing the organization of convection. However, it would not shed as much light on the behaviour of the model, and its ability to produce realistic wind profiles without the constraints of observations.

Mohr and Zipser (1996) stress the importance of the diurnal cycle for MCSs, finding that oceanic MCSs are more numerous at sunrise, whereas continental MCSs are more numerous at sunset. This aspect of the organization of convection was not investigated here, but it would be possible to do so using this method. The fact that we have found differences between those profiles that occur over land or ocean suggests that we might be able to look at different dynamic conditions and see if this has a bearing on the types of MCSs that are formed.

With the ability to identify where shear conditions favourable for the organization of convection occur in a climate model, a logical follow on question is: "how would these conditions affect the CPS of the model?". Several studies have looked at including some of the effects of mesoscale organization into a CPS, e.g. Donner (1993); Alexander and Cotton (1998); Gray (2000). These are based on the empirical studies of Leary and Houze Jr. (1980) in the case of Donner (1993) and Alexander and Cotton (1998), and on the use of high-resolution models of convection in the case of Gray (2000). Furthermore, Mapes

and Neale (2011) provide a means of representing subgrid organization of convection through modifications to the CPS's entrainment rate, which demonstrates another method of modifying a CPS in the presence of organization. Likewise, Moncrieff et al. (2017) modify a CPS to represent the organization of convection under the choice of two arbitrary coefficients which represent the strength of the organization. Discussing Moncrieff et al. (2017), Houze Jr. (2018) notes "These coefficients have the potential of being functions of the large-scale shear, thus making this MCS parameterization consistent with the effects of

shear in controlling MCS dynamics".

Here, we propose that a method similar to that used by Gray (2000) could be used, but based on the RWPs that we have identified. Specifically, the RWPs could be used to drive high-resolution idealized radiative-convective equilibrium experiments in order to induce organization in those experiments. The shear-induced organization of convection would then give an equilibrium response, in a manner similar to that of Robe and Emanuel (2001), which would be suitable for working out how

to modify an equilibrium CPS. The convective response, in terms of heating and moistening terms (Q1 and Q2, e.g. Yanai et al. (1973)), could be diagnosed, and this could be used to inform the design of modifications to the CPS. Furthermore, the strength of the organization could be assessed and used as an input to modifications such as those described in Mapes and Neale (2011) and Moncrieff et al. (2017). The diagnosis of an active RWP in a given grid-column could then be used as additional input to the "shear-aware" CPS, and the results of the high-resolution experiments used to produce a modified CPS to take into account

the effects of shear-induced organization.





## 5 Conclusions

We have developed a procedure for grouping similar wind profiles produced by a climate model into 10 RWPs which effectively span the parameter space. To do this, we made decisions about which wind profiles were important for the organization of convection, and correspondingly we filtered out profiles that did not come from regions where there was significant CAPE, or
that did not have large shears in the lower troposphere. Then, to limit the effect of upper level winds on the analysis, we favour the lower troposphere by reducing the weighting of the winds above 500 hPa. We also rotated the profiles so that their winds at 850 hPa were aligned, as the relative rotation of the profiles is unlikely to have a large effect on the organization that they induce.

To reduce the number of dimensions of our samples, we apply principal component analysis. We use as many principal
components as are required to explain 90 % of the variance, which is seven. Using the samples as represented by their principal components, we apply a K-Means clustering algorithm. This effectively partitions the space of all samples into a given number of groups of profiles. We choose to use 10 clusters, as a pragmatic way of splitting up the profiles so that the spread of each cluster is not too large, and that there is a manageable number of clusters. The median of each cluster then forms our RWPs.

Profiles of the 10 RWPs show that there is not too much spread below 500 hPa. The RWPs exhibit some characteristics that
are associated with the organization of convection, namely low-level and mid-level shear. Their spatial distribution shows that each RWP occurs over preferred regions of the tropics. For example, RWP C6 is well localized to the north-west Pacific. They are well split into oceanic and land profiles, with only one of the RWPs having around 50 % of its profiles coming from both. Some of the RWPs show turning with height. Several show consistent orientation of their 850 hPa winds.

From their seasonal distributions, several RWPs are seen to be most active at two times of the year. For three of the RWPs,
this can be related to being active in one of the hemispheres: C1 shows peak activity in the winter hemisphere, whereas C2 and C10 show peak activity in the summer hemisphere. C6 is seen to be well localized in time over July and August. In general, the RWPs do not show much annual variation.

The distribution of RWPs can be compared with previous studies on the distribution of MCSs. The distribution of RWPs is seen to be broadly consistent with the distribution of MCSs from Mohr and Zipser (1996), and to a lesser extent Tan et al.
(2015); Huang et al. (2018). With Mohr and Zipser (1996), a similar seasonal progression of RWP distribution over Africa is seen as the progression of MCSs. The RWP C6 is seen to be consistent with the higher activity of MCSs over the north-west Pacific. There are some discrepancies though, such as over the Horn of Africa, and over the west Indian Ocean. These may be related to biases in precipitation in the UM over the Indian Ocean, as described in Bush et al. (2015).

The RWPs are similar to profiles observed in studies of organization of convection in specific geographical regions. In
particular, C10 is similar to regional studies from Venezuela (Betts et al., 1976), Brazil (Cohen et al., 1995) and the tropical Pacific (LeMone et al., 1998). C10 was also seen to be similar in form to C2 and C4, which indicates that these RWPs may be associated with the organization of convection as well.

This study builds confidence that a climate model can produce the necessary environmental conditions for shear-induced organization of convection in realistic places. Given that most convection parametrization schemes do not represent the or-



ganization of convection, it indicates regions where modifications to such a scheme could be made in order to address this shortcoming. The question of how to modify a convection parametrization scheme remains; this will be addressed in future studies, although we have set out some suggestions for how this might be achieved (Sect. 4.5).

*Code and data availability.* The UM is available for use under licence. A number of research organisations and national meteorological services use the UM in collaboration with the Met Office to undertake basic atmospheric process research, produce forecasts, develop the

UM code and build and evaluate Earth system models. For further information on how to apply for a licence see https://www.metoffice.gov.uk/research/approach/collaboration/unified-model/partnership.

The data from the UM simulation are available here: http://doi.org/10.6084/m9.figshare.7831574. This includes the $u$ and $v$ winds at 20 pressure levels, and CAPE.

All the analysis code is available online. The analysis code is in the git repository: https://github.com/markmuetz/cosar_analysis, and the

code for running the analysis is available here: https://github.com/markmuetz/omnium. Version 0.8.4 of cosar_analysis (https://github.com/markmuetz/cosar_analysis/archive/v0.8.4.zip) and 0.11.3 of omnium (https://github.com/markmuetz/omnium/archive/v0.11.3.zip) were used to produce all analysis and figures.

*Author contributions.* Contributions by the different authors include conceiving the study (MRM, RSP and PAC), development of all methods and plotting of figures, and writing of the manuscript (MRM), performing the simulations (MRM), and feedback on the analysis as it

progressed and on the manuscript (RSP, PAC, AJS and SJW).

*Competing interests.* The authors declare that there are no competing interests.

*Acknowledgements.* MRM was supported by a SCENARIO/Natural Environment Research Council (NERC) Doctoral Training Partnership award (1642513) as well as by a Met Office CASE studentship. PAC, RSP and SJW were supported by the RevCon (NE/N013743/1) and Circle-A (NE/N013735/1) projects, which are part of the ParaCon research programme. SJW was also supported by the National Centre for

Atmospheric Science, a NERC Collaborative Centre.

The simulation was run using the ARCHER UK National Supercomputing Service (http://www.archer.ac.uk).





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
