# Peer review of "A climatology of tropical wind shear produced by clustering wind profiles from the Met Office Unified Model (GA7.0)"

_Geoscientific Model Development, 2020_

## Referee Comment (RC1)

**Review of "A Climatology of Tropical Wind Shear Produced by Clustering Wind Profiles from a Climate Model" by Muetzelfeldt  et al. (2020)**

The procedure for producing a climatology of tropical wind shear from the UK Met Office Unified Model (UM) is a significant step forward regarding quantification of the physical and dynamical efficacy of global climate models (GCMs). Arguably, it should be a standard diagnostic for GCMs regarding the treatment of moist processes, such as convection and the atmospheric water cycle. It is likely that behavior will vary among GCMs because the unification of observations and models is a significant challenge. Observations, convection-resolving models, and dynamical theory show that vertical wind plays a key role in organized moist convection. But organized convection such as mesoscale convective systems (MCS) are missing from GCMs -- not represented by contemporary convective parameterizations and GCMs have insufficient resolution to adequately simulate them.

I strongly advocate this manuscript, but it has deficiencies that require attention:

1) Distinction between vertical-shear/cold-pool interaction in the tropics and mid-latitudes. Thorpe et al. (1982; TMM) pertains to mid-latitudes where mesoscale downdrafts and cold pools tend to be strong. Cold-pools are much weaker in the tropics (especially over oceans) so the balance mechanism between low-level shear and cold-pool vorticity (Rotunno et al. 1988; RKW) is significantly different (see item 4). It is therefore misleading to cite TMM and RKW in the first sentence of a manuscript pertaining to tropical conditions. See Grant et al. (2020) for more information on such concerns.

2) The remark on line 101 "could lead to the development of shear-aware convective parameterization schemes" is redundant or misleading. That subject has been comprehensively addressed using two world-class GCMs.  Moncrieff et al. (2017; MLB) and Moncrieff (2019) are shear-aware. In MLB approximations of the heat and momentum transport tendencies are based on dynamical models of squall-lines, MCSs and tropical superclusters with particular attention to "slantwise layer overturning" that primarily distinguishes organized systems in shear from ordinary cumulus convection. (See item 5 below for some detailed comments.) When implemented in the atmosphere-only NCAR CAM, slantwise overturning improves the tropical distribution of precipitation, the Madden-Julian Oscillation, and convectively coupled Kelvin waves.  It has even stronger impact when implemented with different shear triggers in Department of Energy (DOE) Exascale Energy Earth System Model (E3SM) atmosphere-only and coupled versions (Chen et al. 2021, in review). A key category of organization involves systems that propagate upshear e.g., squall lines and MCS  (also tropical superclusters  Moncrieff & Klinker 1997) which indicates an interesting  scale-invariance. The upscale effects of upshear propagating tropical systems have been recently quantified in an idealized GCM  (Yang et al. 2019).

3) Section 3.1: There is surprisingly little sign of lower-tropospheric tropical easterly jet-like shear in the Indian Ocean, Maritime Continent, tropical Western Pacific which are primary regions for squall-lines, MCSs, and  superclusters  in association with vertically sheared (tropical easterly jet) environments. This is anticipated to negatively impact the occurrence of MJOs and convectively coupled Kelvin waves in these locations.

4) Section 4.2 Venezuelan squall lines. This application of field-measurements and dynamical model verification strengthens the scientific case. Note that Betts et al. Fig. 6 shows the system-relative inflow is entirely from ahead of the squall-lines for bot updrafts and downdrafts and cool mesoscale downdrafts are replaced by forced ascent adiabatic cooling.. The weak mesoscale inflow from the rear of the squall lines does not reach the Earth's surface. This organized structure is represented by the Moncrieff & Miller (1976) propagating squall-line model, but is not consistent with TMM and RKW ideas.

5) The key role of vertical shear on organized convection is summarized by nonlinear dynamical models, which provide the momentum and heat fluxes for slantwise layer overturning. a) Moncrieff and Green (1972) showed long-lived organized convective overturning in constant vertical shear is controlled by CAPE and inflow available kinetic energy (AKE) supplied by sheared inflow and system propagation. These two categories of energy are interdependent in the form of a convective Richardson number Ri = CAPE/AKE. The vertical shear is strong because Ri ~ 1. The organized convective system slants downshear and mesoscale downdrafts do not exist. b) The Moncrieff and Miller (1976) three-dimensional propagating tropical squall-line model features a third category of energy -- work done by a convectively generated horizontal pressure change across the system. This organization is controlled by the Bernoulli number defined as pressure work divided by AKE. c) TMM mid-latitude squall lines model is controlled by the Bernoulli number and the convective Froude number (proportional to the inverse convective Richardson number). d) The Moncrieff (1992) two-dimensional archetypal model features three categories of organization: wave-like propagation; hydrodynamic limit of TMM; and a system whereby jump-like ascent replaces the mesoscale downdraft.

6) The above classes of model indicate that CAPE and vertical shear are interdependent quantities, and mean-flow conditions exist whereby kinetic energy of system propagation and work done by the horizontal pressure gradient are the principal sources of energy rather than CAPE. Summarizing, strong upper-tropospheric shear is problematic because the convective system slants downshear and downdrafts are suppressed. When the upper-tropospheric shear is weak, the updrafts can slant upshear, so precipitation falls into sub-saturated air and generate mesoscale downdrafts in squall lines and MCSs. It is maximally beneficial when the upper-tropospheric shear reverses direction (i.e., tropical jet-like wind profiles). Families of cumulonimbus then get initiated at the leading edge of the mesoscale downdraft outflow and travel rearward relative to the MCS, a procedure continually repeats and generate the key horizontal pressure gradient (see Lafore and Moncrieff 1989). This optimal category of upshear slantwise overturning is the workhorse of the MLB shear-aware organized convection parameterization scheme.

Signed: Mitchell W. Moncrieff (February 9, 2021)

References:

Chen, C.C., J.H. Richter, C. Liu, M.W. Moncrieff, Q. Tang, W. Lin, S. Xie, and P. J. Rasch, 2021: Effects of organized mesoscale heating on the MJO and precipitation in E3SMv1. JAMES, in review.

Grant, L.D., M.W. Moncrieff, T.P. Lane, and S. C. Van Den Heever, 2020: Shear- parallel tropical convective systems: Importance of cold pools and wind shear. Geophysical Research Letters, 47, e2020GL087720, https://doi.org/10.1029/2020GL087720.

Lafore, J-P. and Moncrieff, M.W, 1989: A Numerical Investigation of the Organization and Interaction of the Convective and Stratiform Regions of Tropical Squall Lines. Journal of the Atmospheric Sciences, 46, 521-544.

Moncrieff, M.W, 2019: Toward a dynamical foundation for organized convection parameterization in GCMs. Geophysical Review Letters, 46, https://doi.org/10.1029/2019GL085316.

Moncrieff, M.W., and M. J. Miller, 1976: The Dynamics and Simulation of Tropical Cumulonimbus and Squall-lines. Quarterly Journal of the Royal Meteorological Society, 102, 373-394.

Moncrieff, M.W, 2019: Toward a dynamical foundation for organized convection parameterization in GCMs. GRL, 46, https://doi.org/10.1029/2019GL085316.

Moncrieff, M. W. and E. Klinker, 1997: Organized convective systems in the tropical western Pacific as a process in general circulation models. Quarterly Journal of the Royal Meteorological Society, 123, 805-828.

Yang, Q., A.J. Majda, and M.W. Moncrieff, 2019: Upscale impact of mesoscale convective systems and its parameterization in an idealized GCM for a MJO analog above the equator. J. Atmos. Sci., 76, 865-892, doi: 10.1175/JAS-D-18-0260.1.

---

## Author Comment (AC1)

**Response to referees' comments on "A Climatology of Tropical Wind Shear Produced by Clustering Wind Profiles from the Met Office Unified Model (GA7.0)"**

Mark Muetzelfeldt (mark.muetzelfeldt@reading.ac.uk)

12/4/2021

**Response to Mitchell Moncrieff – Referee 1**

Many thanks for your comments on our paper – we have sought to clarify and improve it in response to them.

1. We have replaced the citations to Rotunno et al. (1988) and Thorpe et al. (1982) with a citation to Moncrieff and Miller (1976) to make the point about shear being important from theory, and added a short summation of why. We have left in one citation to Rotunno et al. (1988) in Sect. 2.2, with the caveat that its applicability in the tropics is disputed because of weaker tropical cold pools, citing Grant et al. (2018) and Grant et al. (2020). We have removed the second citation to Thorpe et al. (1982) in Sect. 2.2.

2. It is our intention to eventually provide supporting information for schemes such as Moncrieff et al. (2017), and the sentence you highlighted does not reflect this properly. We have amended it to: "could provide supporting information for "shear-aware" CPSs such as Moncrieff et al. (2017) that aim to represent some aspects of shear-induced organization. In particular, the continuation of the work here could be used to provide information about the strength of the response in terms of the low-level shear, as represented by their $\alpha$ parameters.". We have also added a citation to Chen et al. (2021), and stated that the continuation of our work may provide an alternative basis for setting the tunable shear-triggering parameter.

3. This is a good point. We have added a short note to this effect at the end of Sect. 3.2, and linked it to known biases of the UM. In discussion of this point, it got us thinking about how our method could be used more generally in climate models to investigate the link between (lack of) vertical shear and the representation of the MJO and Kelvin waves. This is similar to what you said about the value of the approach for diagnosis in your introductory remarks, and we have added an extra sentence to the abstract and Sect. 4.5 (Extensions) to reflect this perspective.

4. We agreed with these comments, and have added a sentence to this effect in Sect. 4.2.

5. We have added an extra sentence to the first paragraph of the introduction, trying to summarize the salient points for our study.

6. When discussing the relative role of lower- and upper-level shear in Sect. 2.2, we included two sentences making the point you raised here and relating it to our work.

**Response to Anonymous Referee 2**

Thank you for your comments and suggestions for modifications. We reread the paper with your comments in mind, and have streamlined a couple of the sections where our description of the methods was unnecessarily detailed (Sects. 2.5 and 2.6). In describing the arbitrary aspects of our paper, we were trying to not just detail what we did, but why we did it, as we believe this is important for a novel method and we find this useful when reading such papers. However, I (Mark) take your point that some sections were overly detailed, and will endeavour to improve my writing style going forward.

Regarding the abstract, your version is a good template for how to condense it whilst retaining most of the salient points. We have used it to rewrite the abstract in our own words, and shorten it (428 to 354 words). We have kept the final short paragraph on a potential extension, as this was motivation for our paper and we believe it should be there. Additionally, we have minorly extended it in light of Referee 1's comments.

Regarding the "representative" hodographs we used the median as the RWP because it is less prone to outliers than other summary measures. We looked at specific wind profiles that were closest to the given RWPs, as per your suggestion, and it was possible to find several that matched each RWP very closely (Fig. 1), even for one of the RWPs with less shear (C4). The overall spread of the 10 closest profiles is small, apart from higher in the atmosphere due to the reduced weighting at those levels. The filtering and normalization should preserve *relative* shears between different levels, and this is what we wanted to obtain from the space of all wind profiles in the clustering algorithm. Some evidence for the success of this is provided by looking at the RWPs and their spread, which clearly still represent strong shear in certain cases (RWPs C2, C6 and C10).

[Figure]

Figure 1: Four RWPs discussed at length in the paper and their 10 closest wind profiles, in order of maximum to minimum low-level shear (a – d). Shows $u'$ (blue) and $v'$ (red) for each RWP (solid) and for the 10 closest wind profiles (dotted).

**Response to Editor Astrid Kerkweg**

We have included the name and version number of the climate model in the title, and have added it to the revised abstract. In the abstract, we have made it clear that the UM is being used as a climate model.

**References**

Chen, C. C., Richter, J. H., Liu, C., Moncrieff, M. W., Tang, Q., Lin, W., Xie, S., and Rasch, P. J.: Effects of organized mesoscale heating on the MJO and precipitation in E3SMv1, Journal of Advances in Modeling Earth Systems (in review), 2021.

Grant, L. D., Lane, T. P., and van den Heever, S. C.: The role of cold pools in tropical oceanic convective systems, Journal of the Atmospheric Sciences, 75, 2615–2634, https://doi.org/10.1175/JAS-D-17-0352.1, 2018.

Grant, L. D., Moncrieff, M. W., Lane, T. P., and van den Heever, S. C.: Shear-parallel tropical convective systems: Importance of cold pools and wind shear, Geophysical Research Letters, 47, e2020GL087 720, https://doi.org/10.1029/2020GL087720, 2020.

Moncrieff, M. W. and Miller, M. J.: The dynamics and simulation of tropical cumulonimbus and squall lines, Quarterly Journal of the Royal Meteorological Society, 102, 373–394, https://doi.org/10.1002/qj.49710243208, 1976.

Moncrieff, M. W., Liu, C., and Bogenschutz, P.: Simulation, modeling, and dynamically based parameterization of organized tropical convection for global climate models, Journal of the Atmospheric Sciences, 74, 1363–1380, https://doi.org/10.1175/JAS-D-16-0166.1, 2017.

Rotunno, R., Klemp, J. B., and Weisman, M. L.: A Theory for Strong, Long-Lived Squall Lines, Journal of the Atmospheric Sciences, 45, 463–485, https://doi.org/10.1175/1520-0469(1988)045⟨0463:ATFSLL⟩2.0.CO;2, 1988.

Thorpe, A. J., Miller, M. J., and Moncrieff, M. W.: Two-dimensional convection in non-constant shear: A model of mid-latitude squall lines, Quarterly Journal of the Royal Meteorological Society, 108, 739–762, https://doi.org/10.1002/qj.49710845802, 1982.